DATA RELEASE

# Genome assembly of the rare and endangered Grantham's camellia, *Camellia granthamiana*

Hong Kong Biodiversity Genomics Consortium*,†

## ABSTRACT

Grantham's camellia (*Camellia granthamiana* Sealy) is a rare and endangered tea species discovered in Hong Kong in 1955 and endemic to southern China. Despite its high conservation value, the genomic resources of *C. granthamiana* are limited. Here, we present a chromosome-scale draft genome of the tetraploid *C. granthamiana* ($2n = 4x = 60$), combining PacBio long-read sequencing and Omni-C data. The assembled genome size is ~2.4 Gb, with most sequences anchored to 15 pseudochromosomes resembling a monoploid genome. The genome has high contiguity, with a scaffold N50 of 139.7 Mb, and high completeness (97.8% BUSCO score). Our gene model prediction resulted in 68,032 protein-coding genes (BUSCO score of 90.9%). We annotated 1.65 Gb of repeat content (68.48% of the genome). Our Grantham's camellia genome assembly is a valuable resource for investigating Grantham's camellia's biology, ecology, and phylogenomic relationships with other *Camellia* species, and provides a foundation for further conservation measures.

**Subjects** Genetics and Genomics, Botany, Plant Genetics

**Submitted:** 11 January 2024

\* Correspondence on behalf of the consortium: E-mail: jeromehui@cuhk.edu.hk

† Collaborative Authors: Entomological experts who validated the dataset and their affiliations appears at the end of the document

Preprint submitted at https://doi.org/10.1101/2024.01.15.575486

Included in the series: ***Hong Kong Biodiversity Genomics*** (https://doi.org/10.46471/GIGABYTE_SERIES_0006)

## INTRODUCTION

*Camellia* is a large genus in the family Theaceae with more than 230 described species [1]. Camellias are well-known for their ornamental and economic values as tea and woody-oil producing plants, with tens of thousands of cultivars derived from them [2]; however, more than 60 *Camellia* species are regarded as globally threatened due to their natural habitat fragmentation or loss, and to their small population size [3]. The Grantham's camellia (*Camellia granthamiana*) (Figure 1A) is a rare species first discovered in Hong Kong and named after the former Governor Sir Alexander Grantham, and is narrowly distributed in Hong Kong and Guangdong, China [3]. It is listed as vulnerable in the Red List of the International Union for Conservation of Nature and recorded as endangered in the China Plant Red Data Book [4]. In Hong Kong, Grantham's camellia is a protected species by law and has been actively propagated and reintroduced to the wild by the Agriculture, Fisheries and Conservation Department [5].

## CONTEXT

In view of the high conservation value of Grantham's camellia, several molecular studies have been done. They included sequencing the chloroplast genomes of *C. granthamiana* [6, 7], using pan-transcriptomes to reconstruct the phylogeny of over a hundred *Camellia* species [8], and population genetics studies [9]. However, the nuclear genomic resources of *C. granthamiana* are still missing. While most *Camellia* species possess a karyotype of $2n = 30$, *C. granthamiana* is an exception with a karyotype of $2n = 4x = 60$ [10, 11].

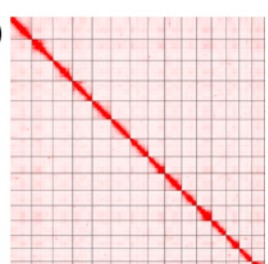
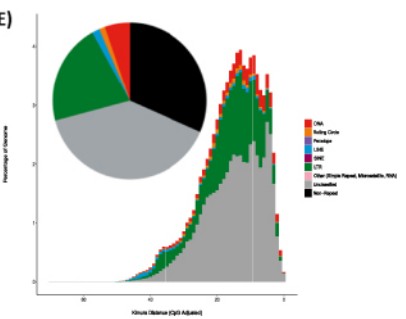

| | *Camellia granthamiana* |
|---|---|
| Genome size (bp) | 2,412,502,632 |
| Number of scaffolds | 1,681 |
| N_count | 0.05% |
| N50 (bp) | 139,717,271 |
| N50n | 8 |
| BUSCO (Genome) | 97.80% |
| Gene models | 62,113 |
| Protein-coding genes | 68,032 |
| BUSCO (Proteome) | 90.90% |

| Chr Number | scaffld_length | scaffld_id | % of whole genome |
|---|---|---|---|
| 1 | 187,610,956 | scaffld_1_1 | 7.78% |
| 2 | 178,682,930 | scaffld_2_1 | 7.41% |
| 3 | 166,889,417 | scaffld_3_1 | 6.92% |
| 4 | 162,452,925 | scaffld_4_1 | 6.73% |
| 5 | 161,471,252 | scaffld_5_1 | 6.69% |
| 6 | 157,427,254 | scaffld_6_1 | 6.53% |
| 7 | 152,269,496 | scaffld_7_1 | 6.31% |
| 8 | 139,717,271 | scaffld_8_1 | 5.79% |
| 9 | 138,255,011 | scaffld_9_1 | 5.73% |
| 10 | 127,884,699 | scaffld_10_1 | 5.30% |
| 11 | 124,989,205 | scaffld_11_1 | 5.18% |
| 12 | 120,464,456 | scaffld_12_1 | 4.99% |
| 13 | 109,020,227 | scaffld_13_1 | 4.52% |
| 14 | 108,330,513 | scaffld_14_1 | 4.49% |
| 15 | 103,949,816 | scaffld_15_1 | 4.31% |
| SUM: | 2,139,415,428 | | 88.68% |
| BUSCO: | 96.0%[S:77.4%,D:18.6%] | | |

**Figure 1.** Genomic information of *Camellia granthamiana*. (A) Picture of *Camellia granthamiana*; (B) Summary of genome statistics; (C) Omni-C contact map of the genome assembly; (D) Information of 15 pseudochromosomes; (E) Pie chart (Top) and repeat landscape plot (bottom) of repetitive elements in the genome.

In Hong Kong, *C. granthamiana* was chosen as one of the species listed for sequencing in the Hong Kong Biodiversity Genomics Consortium (also known as EarthBioGenome Project Hong Kong), which is formed by investigators from eight publicly funded universities. Here, we report the genome assembly of *C. granthamiana*, which can serve as a solid foundation for further investigations of this rare and endangered species.

## METHODS

### Sample collection and high molecular weight DNA extraction

Fresh leaf tissues were sampled in transplanted individuals on the campus of the Chinese University of Hong Kong. High molecular weight (HMW) genomic DNA was isolated from 1 g leaf tissues using pretreatment with cetyltrimethylammonium bromide (CTAB) followed by the NucleoBond HMW DNA kit (Macherey Nagel Item No. 740160.20). Briefly, tissues were ground with liquid nitrogen and digested in 5 mL CTAB buffer [12] with the addition of 1% polyvinylpyrrolidone for 1 h. The lysate was treated with RNAse A, followed by the addition of 1.6 mL of 3 M potassium acetate and two rounds of chloroform:IAA (24:1) washes. The supernatant was transferred to a new 50 mL tube using a wide-bore tip. H1 buffer from the NucleoBond HMW DNA kit was added to the supernatant for a total volume of 6 mL, from which the DNA was isolated following the manufacturer's protocol. After the DNA was eluted with 60 µL elution buffer (PacBio Ref. No. 101-633-500), a quality check was carried out with NanoDrop™ One/OneC Microvolume UV–Vis Spectrophotometer, Qubit® Fluorometer, and overnight pulse-field gel electrophoresis.

### Pacbio library preparation and sequencing

The qualified DNA was sheared with a g-tube (Covaris Part No. 520079) with six passes of centrifugation at 1,990 × *g* for 2 min. Next, it was purified with SMRTbell® cleanup beads

**Table 1.** Genome and transcriptome sequencing information.

| Library | Reads | Bases | Accession number |
|---|---|---|---|
| PacBio HiFi | 5,071,365 | 54,421,045,547 | SRR26895683 |
| Omnic | 1,558,845,532 | 233,826,829,800 | SRR26909376 |
| mRNA CamG_YL_H | 41,179,150 | 5,987,388,039 | SAMN40925022 |

(PacBio Ref. No. 102158-300). A total of 2 µL sheared DNA was taken for fragment size examination through overnight pulse-field gel electrophoresis. Then, two SMRTbell libraries were constructed with the SMRTbell® prep kit 3.0 (PacBio Ref. No. 102-141-700) following the manufacturer's protocol. The final library was prepared with the Sequel® II binding kit 3.2 (PacBio Ref. No. 102-194-100) and was loaded, using the diffusion loading mode, with the on-plate concentration set at 90 pM on the Pacific Biosciences SEQUEL IIe System, running for 30-hour movies to output HiFi reads. In total, three SMRT cells were used for the sequencing. Details of the resulting sequencing data are summarized in Table 1.

### Omni-C library preparation and sequencing

Nuclei were isolated from 3 g fresh leaf tissues ground with liquid nitrogen using the PacBio protocol modified from Workman *et al.* [13]. The nuclei pellet was snap-frozen with liquid nitrogen and stored at −80 °C. Upon Omni-C library construction, the nuclei pellet was resuspended in 4 mL 1× PBS buffer and processed with the Dovetail® Omni-C® Library Preparation Kit (Dovetail Cat. No. 21005) following the manufacturer's procedures. The concentration and fragment size of the resulting library were assessed by Qubit® Fluorometer and TapeStation D5000 HS ScreenTape, respectively. The qualified library was sent to Novogene and sequenced on an Illumina HiSeq-PE150 platform. Details of the resulting sequencing data are summarized in Table 1.

### Total RNA isolation and transcriptome sequencing

Approximately 0.5 g of young leaf tissue was ground into powder after being frozen in liquid nitrogen. Total RNA was then isolated using a CTAB pretreatment method [14], followed by the mirVana miRNA Isolation Kit (Ambion, cat no. AM1560). The quality of the RNA sample was assessed using NanoDrop® One/OneC Microvolume UV–Vis Spectrophotometer and 1% agarose gel electrophoresis. Next, the sample was sent to Novogene Co. Ltd (Hong Kong, China) for transcriptome sequencing. Details of the sequencing data are listed in Table 1.

### Genome assembly and gene model prediction

*De novo* genome assembly was first generated with Hifiasm (RRID:SCR_021069) [15] and then was processed by searching it against the NT database with BLASTn (RRID:SCR_004870) to remove possible contaminations using Blobtools (v1.1.1; RRID:SCR_017618) [16]. Subsequently, haplotypic duplications were removed according to the depth of HiFi reads using purge_dups (RRID:SCR_021173) [17]. Proximity ligation data from Omni-C were used to scaffold the assembly with YaHS (RRID:SCR_022965) [18].

To remove low-quality and contaminated reads, RNA sequencing data were first processed using Trimmomatic (v0.39; RRID:SCR_011848) [19], with parameters "TruSeq3-PE.fa:2:30:10 SLIDINGWINDOW:4:5 LEADING:5 TRAILING:5 MINLEN:25" [19], and kraken2 (v2. 0.8 with kraken2 database k2_standard_20210517; RRID:SCR_005484) [20].

Then, RNA sequencing data were aligned to the repeat soft-masked genome using Hisat2 (RRID:SCR_015530) [21] to generate the bam file. A total of 6,219,463 Tracheophyta reference protein sequences were downloaded from NCBI as protein hits, along with the RNA bam file, to perform genome annotation using Braker (v3.0.8; RRID:SCR_018964) [22] with default parameters.

### Repeat annotation
The annotation of transposable elements (TEs) was performed by the Earl Grey TE annotation pipeline (version 1.2) [23].

### Macrosynteny analysis
The longest gene transcripts from the predicted gene models of *C. granthamiana* and *Camellia sinensis* (accession number: GWHASIV00000000) [24] were used to retrieve orthologous gene pairs with reciprocal BLASTp (e-value 1e-5; RRID:SCR_001010) using diamond (v2.0.13; RRID:SCR_016071) [25]. The BLAST output was passed to MCScanX (RRID:SCR_022067) [26] to infer the macrosynteny of the pseudochromosomes between *C. granthamiana* and *C. sinensis* with default parameters.

### DATA VALIDATION AND QUALITY CONTROL
For the HMW DNA and Pacbio library samples, NanoDrop® One/OneC Microvolume UV–Vis Spectrophotometer, Qubit® Fluorometer, and overnight pulse-field gel electrophoresis were used for quality control. The quality of the Omni-C library was checked by Qubit® Fluorometer and TapeStation D5000 HS ScreenTape. Hi-C contact maps used to validate the pseudochromosomes were generated using the Juicer tools (version 1.22.01; RRID:SCR_017226) [27], following the Omni-C manual (Figure 1C) [28].

During genome assembly, BlobTools (v1.1.1) [16] was used to remove possible contaminations (Figure 3). The resulting genome assembly was run with BUSCO v5.5.0 [29], using the Viridiplantae dataset (Viridiplantae Odb10) to assess the completeness of the genome assembly and gene annotation.

Omni-C reads and PacBio HiFi reads were used to measure the assembly completeness and the consensus quality (QV) using Merqury (v1.3; RRID:SCR_022964) [30] with kmer 21, resulting in a 95.7267% kmer completeness for the Omni-C data and 52.3372 QV values for the HiFi reads, corresponding to 99.999% accuracy.

### RESULTS AND DISCUSSION
### Genome assembly of *C. granthamiana*
A total of 54.4 Gb HiFi reads was yielded from PacBio sequencing with an average length of 10,731 bp (Tables 1, 2). Together with 233.8 Gb Omni-C data, the genome of *C. granthamiana* was assembled to a final size of 2,412.5 Mb with 6,572 gaps and 37.64% GC content, from which 88.68% of the sequences were anchored into 15 pseudochromosomes (Figure 1B–D). The scaffold N50 was 139.7 Mb and the BUSCO score (RRID:SCR_015008) was 97.8% (Figure 1B; Table 2). Our gene model prediction yielded a total of 68,032 protein-coding genes with a mean length of 298 amino acids and a BUSCO score of 90.9%, which is comparable to other *Camellia* species (Tables 3, 4).

Repeat content analysis annotated 1.65 Gb of transposable elements (TEs), comprising 68.48% of the *C. granthamiana* genome. Among the classified TEs, long terminal repeats

**Table 2.** Genome statistics and sequencing information.

| | *Camellia granthamiana* |
|---|---|
| Total length (bp) | 2,412,502,632 |
| number | 1,681 |
| Mean length (bp) | 1,435,159 |
| Longest | 187,610,956 |
| Shortest | 1,000 |
| N_count | 0.054% |
| Gaps | 6,572 |
| N50 | 139,717,271 |
| N50n | 8 |
| N70 | 124,989,205 |
| N70n | 11 |
| N90 | 1,975,528 |
| N90n | 27 |
| BUSCOs (Genome) | C:97.9%[S:79.3%,D:18.6%],F:0.5%,M:1.6%,n:425 |
| HiFi Reads | 5,071,365 |
| HiFi Bases | 54,421,045,547 |
| HiFi Q30% | 3 |
| HiFi Q20% | 5 |
| HiFi GC% | 38 |
| HiFi Nppm | 0 |
| HiFi Ave_len | 10,731 |
| HiFi Min_len | 100 |
| HiFi Max_len | 50,499 |
| Gene models | 74,088 |
| No. of protein-coding genes | 76,992 |
| Total length of protein-coding genes (AA) | 23,158,643 |
| Mean length of protein-coding genes (AA) | 301 |
| BUSCOs (Proteome) | C:85.9%[S:66.1%,D:19.8%],F:8.5%,M:5.6%,n:425 |

**Table 3.** Camellia genome statistics.

| Species | Assembly accession | BUSCOs (Genome) | genome_size (bp) | N50 |
|---|---|---|---|---|
| *Camellia sinensis* var. sinensis | GCA_004153795.2 | C:87.8%[S:77.9%,D:9.9%],F:4.9%,M:7.3%,n:425 | 2,863,254,423 | 1,320,966 |
| *Camellia sinensis* | GCA_013676235.1 | C:97.9%[S:88.7%,D:9.2%],F:0.9%,M:1.2%,n:425 | 3,113,463,150 | 204,241,410 |
| *Camellia sinensis* | GCF_004153795.1 | C:94.4%[S:83.3%,D:11.1%],F:3.5%,M:2.1%,n:425 | 3,105,370,065 | 1,388,941 |
| *Camellia lanceoleosa* | GCA_025200525.1 | C:98.8%[S:80.9%,D:17.9%],F:0.7%,M:0.5%,n:425 | 2,999,357,698 | 186,426,707 |
| **Camellia granthamiana** | **GCA_036172215.1** | **C:97.9%[S:79.3%,D:18.6%],F:0.5%,M:1.6%,n:425** | **2,412,502,632** | **139,717,271** |
| *Camellia sinensis* var. sinensis | GCA_017311205.1 | C:97.4%[S:85.9%,D:11.5%],F:0.7%,M:1.9%,n:425 | 3,062,881,361 | 213,467,978 |
| *Camellia sinensis* var. sinensis | GCA_020536495.1 | C:97.1%[S:84.9%,D:12.2%],F:1.4%,M:1.5%,n:425 | 3,062,744,301 | 213,458,217 |
| *Camellia sinensis* | GCA_020536515.1 | C:97.2%[S:85.4%,D:11.8%],F:0.9%,M:1.9%,n:425 | 3,062,857,199 | 213,466,203 |
| *Camellia sinensis* var. lasiocalyx | GCA_020536555.1 | C:97.4%[S:85.4%,D:12.0%],F:0.9%,M:1.7%,n:425 | 3,062,765,809 | 213,459,538 |
| *Camellia sinensis* var. assamica | GCA_020536565.1 | C:97.4%[S:85.9%,D:11.5%],F:0.7%,M:1.9%,n:425 | 3,062,795,309 | 213,462,283 |
| *Camellia sinensis* | GCA_020536595.1 | C:97.4%[S:85.6%,D:11.8%],F:0.7%,M:1.9%,n:425 | 3,062,747,348 | 213,457,662 |
| *Camellia sinensis* var. assamica | GCA_020536795.1 | C:97.4%[S:85.9%,D:11.5%],F:0.9%,M:1.7%,n:425 | 3,062,621,441 | 213,448,988 |
| *Camellia sinensis* var. assamica | GCA_020536855.1 | C:97.1%[S:84.9%,D:12.2%],F:1.2%,M:1.7%,n:425 | 3,062,795,300 | 213,461,895 |
| *Camellia sinensis* var. assamica | GCA_020536865.1 | C:97.4%[S:86.1%,D:11.3%],F:0.9%,M:1.7%,n:425 | 3,062,765,203 | 213,459,320 |
| *Camellia oleifera* | GCA_022316695.1 | C:96.0%[S:69.6%,D:26.4%],F:0.7%,M:3.3%,n:425 | 2,889,508,820 | 185,364,083 |
| *Camellia japonica* | GCA_030407325.1 | C:97.9%[S:78.6%,D:19.3%],F:0.2%,M:1.9%,n:425 | 2,803,480,011 | 175,506,177 |
| *Camellia sinensis* | GCA_032173705.1 | C:88.4%[S:79.5%,D:8.9%],F:7.5%,M:4.1%,n:425 | 2,679,620,955 | 146,057,547 |

retrotransposons accounted for the largest proportion (20.99%), followed by DNA transposons (5.30%), long interspersed nuclear elements (1.60%), and rolling-circle transposons (1.21%) (Figure 1D; Table 5). The large proportion of repeat content in the *C. granthamiana* genome is comparable to other tea species, such as the Tieguanyin cultivar

**Table 4.**  Camellia genome annotations statistics.

| Species | *Camellia sinensis* var. *sinensis* | *Camellia sinensis* | *Camellia sinensis* | *Camellia lanceoleosa* | *Camellia granthamiana* |
|---|---|---|---|---|---|
| Assembly Accession | GCA_004153795.2 | GCA_013676235.1 | GCF_004153795.1 | GCA_025200525.1 | **GCA_036172215.1** |
| genome_size(bp) | 2,863,254,423 | 3,113,463,150 | 3,105,370,065 | 2,999,357,698 | **2,412,502,632** |
| BUSCO (Prot) | C:71.6% [S:64.5%,D:7.1%], F:12.7%, M:15.7%, n:425 | C:83.1% [S:80.7%,D:2.4%], F:8.5%, M:8.4%, n:425 | C:96.5% [S:44.5%,D:52.0%], F:1.9%, M:1.6%, n:425 | C:95.0% [S:81.6%,D:13.4%], F:3.8%, M:1.2%, n:425 | **C:90.9% [S:68.5%,D:22.4%], F:3.1%, M:6.0%, n:425** |
| Number_of_Proteins | 30,173 | 32,356 | 76,698 | 54,167 | **68,032** |
| Sum_of_Amino_Acids (aa) | 13,483,688 | 12,149,973 | 31,189,428 | 18,299,499 | **20,299,147** |
| Mean_of_Proteins (aa) | 447 | 376 | 407 | 338 | **298** |
| Sum_of_Exons (bp) | 49,451,194 | 36,546,930 | 176,395,894 | 55,437,744 | **60,897,420** |
| Mean_of_Exons (bp) | 285 | 213 | 294 | 218 | **267** |
| Sum_of_Introns (bp) | 170,601,000 | 187,621,926 | 1,233,047,059 | 335,774,258 | **196,881,812** |
| Mean_of_Introns (bp) | 1,191 | 1,307 | 2,521 | 1,674 | **1,225** |
| Numer_of_gene_loci | 30,173 | 32,356 | 62,338 | 54,167 | **62,113** |
| Sum_of_gene_region_(bp) | 220,052,194 | 255,832,614 | 355,988,266 | 390,833,969 | **218,557,015** |
| %_of_gene_loci_in_genome | 7.69% | 8.22% | 11.46% | 13.03% | **9.06%** |
| Average_gene_region(bp) | 7,293 | 7,907 | 5,711 | 7,215 | **3,519** |

**Table 5.**  Summary of the classified TEs in the genome.

| Classification | Total length (bp) | Count | Proportion (%) | No. of distinct classifications |
|---|---|---|---|---|
| DNA | 127,797,240 | 123,181 | 5.30 | 7,315 |
| LINE | 38,512,116 | 34,251 | 1.60 | 5,403 |
| LTR | 506,291,722 | 183,247 | 20.99 | 8,722 |
| Other (Simple Repeat, Microsatellite, RNA) | 1,176,745 | 2,020 | 0.05 | 598 |
| Penelope | 138,228 | 226 | 0.01 | 131 |
| Rolling Circle | 29,291,650 | 25,351 | 1.21 | 3,413 |
| SINE | 421,211 | 1,239 | 0.02 | 285 |
| Unclassified | 948,396,501 | 917,104 | 39.31 | 9,029 |
| **SUM:** | **1,652,025,413** | **1,286,619** | **68.48** | **34,896** |

of *C. sinensis* (78.2%) [24], wild oil-Camellia *Camellia oleifera* (76.1%) [31], and *Camellia chekiangoleosa* (79.09%) [32].

## Macrosynteny between *C. granthamiana* and *C. sinensis*

Our macrosynteny analysis revealed a 1-to-1 pair relationship between the 15 pseudochromsomes of *C. granthamiana* and *C. sinensis* (Figure 2). This indicates that the assembled 15 pseudochromosomes resemble a monoploid genome of the tetraploid *C. granthamiana*.

## CONCLUSION AND FUTURE PERSPECTIVES

This study presents the first *de novo* genome assembly of the rare and endangered *C. granthamiana*. This valuable genome resource has excellent potential for use in future studies on the conservation biology of Grantham's camellia, its relationship with other Camellia species from a phylogenomic perspective, and further investigations on the biosynthesis of secondary metabolites in tea species.

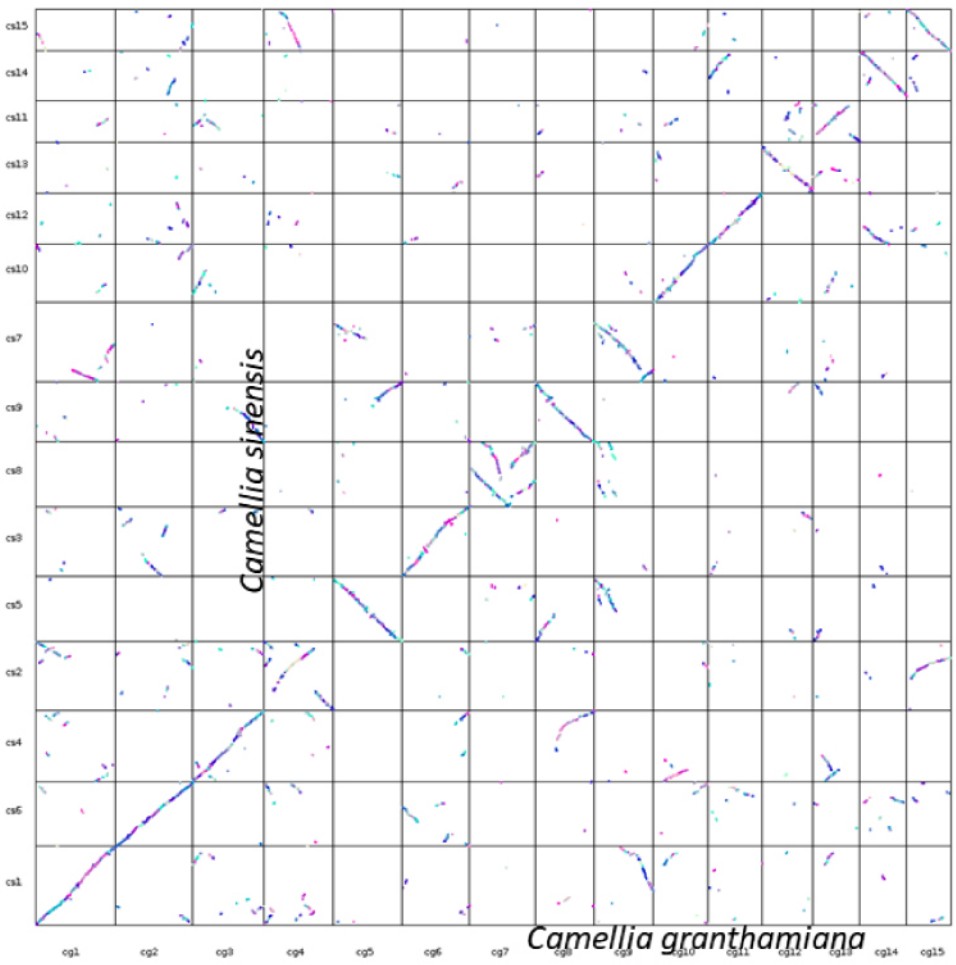

**Figure 2.** Macrosynteny dot plot between *Camellia granthamiana* and *Camellia sinensis*.

## DISCLAIMER

The genomic data generated in this study was not fully haplotype-resolved for a tetraploid genome, and the genome heterozygosity was not assessed.

## DATA AVAILABILITY

The final genome assembly of this study was submitted to NCBI under the accession number JAXFYN000000000. The generated raw reads were deposited in the NCBI database under the SRA accessions SRR26895683, SRR26909376, and SAMN40925022. The genome annotations and other supporting data files are available in Figshare [33].

## ABBREVIATIONS

CTAB, cetyl trimethylammonium bromide; HMW, high molecular weight; QV, consensus quality; TE, transposable elements.



**Figure 3.** Genome assembly quality control and contaminant/cobiont detection.

## DECLARATIONS

## Ethics approval and consent to participate

The authors declare that ethical approval was not required for this type of research.

## Competing interests

The authors declare that they do not have competing interests.

## Authors' contribution

JHLH, TFC, LLC, SGC, CCC, JKHF, JDG, SCKL, YHS, CKCW, KYLY and YW conceived and supervised the study. DTWL collected the sample materials. STSL and WLS performed DNA extraction, library preparation and genome sequencing. HYY facilitated the logistics of samples. WN performed genome assembly, gene model prediction and genome quality check analyses. STSL carried out the macrosynteny analysis.

## Funding

This work was funded and supported by the Hong Kong Research Grant Council Collaborative Research Fund (C4015-20EF), CUHK Strategic Seed Funding for Collaborative Research Scheme (3133356), and CUHK Group Research Scheme (3110154).

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

## DETAILS OF COLLABORATIVE AUTHORS

### • List of authors in Hong Kong Biodiversity Genomics Consortium

Jerome H. L. Hui,[1] Ting Fung Chan,[2] Leo Lai Chan,[3] Siu Gin Cheung,[4] Chi Chiu Cheang,[5,6] James Kar-Hei Fang,[7] Juan Diego Gaitan-Espitia,[8] Stanley Chun Kwan Lau,[9] Yik Hei Sung,[10,11] Chris Kong Chu Wong,[12] Kevin Yuk-Lap Yip,[13,14] Yingying Wei,[15] Sean Tsz Sum Law,[1] Wai Lok So,[1] Wenyan Nong,[1] David Tai Wai Lau,[16] Ho Yin Yip[1]

[1]School of Life Sciences, Simon F.S. Li Marine Science Laboratory, State Key Laboratory of Agrobiotechnology, Institute of Environment, Energy and Sustainability, The Chinese University of Hong Kong, Hong Kong, China

[2]School of Life Sciences, State Key Laboratory of Agrobiotechnology, The Chinese University of Hong Kong, Hong Kong SAR, China

[3]State Key Laboratory of Marine Pollution and Department of Biomedical Sciences, City University of Hong Kong, Hong Kong SAR, China

[4]State Key Laboratory of Marine Pollution and Department of Chemistry, City University of Hong Kong, Hong Kong SAR, China

[5]Department of Science and Environmental Studies, The Education University of Hong Kong, Hong Kong SAR, China

[6]EcoEdu PEI, Charlottetown, PE, C1A 4B7, Canada

[7]Department of Food Science and Nutrition, Research Institute for Future Food, and State Key Laboratory of Marine Pollution, The Hong Kong Polytechnic University, Hong Kong SAR, China

[8]The Swire Institute of Marine Science and School of Biological Sciences, The University of Hong Kong, Hong Kong SAR, China

[9]Department of Ocean Science, The Hong Kong University of Science and Technology, Hong Kong SAR, China

[10]Science Unit, Lingnan University, Hong Kong SAR, China

[11]School of Allied Health Sciences, University of Suffolk, Ipswich, IP4 1QJ, UK

[12]Croucher Institute for Environmental Sciences, and Department of Biology, Hong Kong Baptist University, Hong Kong SAR, China

[13]Department of Computer Science and Engineering, The Chinese University of Hong Kong, Hong Kong SAR, China

[14]Sanford Burnham Prebys Medical Discovery Institute, La Jolla, CA, USA

[15]Department of Statistics, The Chinese University of Hong Kong, Hong Kong SAR, China

[16]Shiu-Ying Hu Herbarium, School of Life Sciences, The Chinese University of Hong Kong, Hong Kong SAR, China

