## [Reviewer Report]

Comments on revised manuscriptThe authors have replied to most of the comments reasonably. The manuscript is now clearly written.  Line 93, " in transplanted individual", should be in "a" or "the" transplanted individual.

---

## [Editor Report]

Editor’s AssessmentThis work is part of a series of papers from the Hong Kong Biodiversity Genomics Consortium sequencing the rich biodiversity of species in Hong Kong. This example assembles the genome of Grantham's camellia (Camellia granthamiana) a rare and endangered tea species discovered in Hong Kong and narrowly distributed in Southern China. In view of the high conservation value of Grantham's camellia a reference genome was sequenced. This data was produced using PacBio HiFi reads and Omni-C sequencing data, the resulting genome assembly being around 2.4Gb in size. From this 68,032 protein-coding genes were predicted. After review improved the methodological details and added new RNA-seq data the quality metrics look near chromosome-level, having a scaffold N50 of 139.7 Mb, and high completeness (97.8% BUSCO). This high-quality genome should hopefully be a valuable resource for use in future studies on the conservation biology of Grantham's camellia, its phylogenomic relationship with other Camellia species, and further investigations on the biosynthesis of secondary metabolites in tea.

---

## [Reviewer Report]

Reviewer name and names of any other individual's who aided in reviewer Yongpeng MaDo you understand and agree to our policy of having open and named reviews, and having your review included with the published papers. (If no, please inform the editor that you cannot review this manuscript.)YesIs the language of sufficient quality?YesPlease add additional comments on language quality to clarify if needed
Are all data available and do they match the descriptions in the paper? YesAdditional Comments1) Please provide the number of gaps and GC content of the genome. 2) A new table should be added to explain the annotations, including the number of genes, the number of cds, the number of introns, etc. 3) The content of different types of repeating elements in the genome needs to be shown (e.g. LTR, copia, gypsy, line, sine, TIR, etc.) 4) line 141, was there a process of quality filtering of the RNA data in this study?  5) Were chromosomes named according to haplotype? 6) line 185，figure2 was missing in the pdf version. 7）figure1 has no units for all parameters except genome size. 8) The total number of protein-coding genes (76992) could be overestimated evidenced by lower BUSCO score, probably due to very limited samples for RNA-seq. Are the data and metadata consistent with relevant minimum information or reporting standards? See GigaDB checklists for examples <a href="http://gigadb.org/site/guide" target="_blank">http://gigadb.org/site/guide</a>NoAdditional CommentsIs the data acquisition clear, complete and methodologically sound?YesAdditional CommentsIs there sufficient detail in the methods and data-processing steps to allow reproduction?YesAdditional CommentsIs there sufficient data validation and statistical analyses of data quality? YesAdditional CommentsIs the validation suitable for this type of data?YesAdditional CommentsIs there sufficient information for others to reuse this dataset or integrate it with other data?YesAdditional CommentsAny Additional Overall Comments to the AuthorRecommendationMajor Revision

---

## [Reviewer Report]

Reviewer name and names of any other individual's who aided in reviewer Fu-Sheng YangDo you understand and agree to our policy of having open and named reviews, and having your review included with the published papers. (If no, please inform the editor that you cannot review this manuscript.)YesIs the language of sufficient quality?YesPlease add additional comments on language quality to clarify if needed
Are all data available and do they match the descriptions in the paper? NoAdditional CommentsAre the data and metadata consistent with relevant minimum information or reporting standards? See GigaDB checklists for examples <a href="http://gigadb.org/site/guide" target="_blank">http://gigadb.org/site/guide</a>YesAdditional CommentsIs the data acquisition clear, complete and methodologically sound?NoAdditional CommentsIs there sufficient detail in the methods and data-processing steps to allow reproduction?NoAdditional CommentsIs there sufficient data validation and statistical analyses of data quality? NoAdditional CommentsIs the validation suitable for this type of data?NoAdditional CommentsIs there sufficient information for others to reuse this dataset or integrate it with other data?NoAdditional CommentsAny Additional Overall Comments to the AuthorThe authors present a chromosome-scale draft genome of the tetraploid Camella granthamiana using a combination of PacBio long read sequencing and Omni-C data. The assembled sequences were anchored to 15 pseudochromosomes that resemble a monoploid genome. The genome is of high contiguity, with a scaffold N50 of 139.7 Mb, and a high completeness with a 97.8% BUSCO score. Gene model prediction resulted in a total 76,992 protein-coding genes. While the paper is well written, I have some comments and suggestions.  1. I noticed that the Camella species is a tetraploid species with 60 chromosomes. The authors resembled the sequences to 15 pseudochromosomes and generated a mosaic assembly of two, or more haplotypes. I am concerned about the assembly quality of the sequences, in the absence of a paternal reference genome. I suggest that some special assembly techniques are necessary to ensure the assemble quality.  2. The authors used public RNA sequences to annotate the de novo genome. I do not believe this is a normal and good method. I suggest the authors to sequence RNAs from different tissues from the transplanted individual on the campus of the Chinese University of Hong Kong，and use these new RNA sequences from the same individuals to annotate the genome. 3. To date, more and more genomes of Camella have been sequenced. I suggest the authors compare the genome of Camella granthamiana with other Camella species, and try to find the potential parents, and reveal the potential mechanism of rare and endangered. 
RecommendationMajor Revision